# Characterization of Two Complete Mitochondrial Genomes of Ledrinae (Hemiptera: Cicadellidae) and Phylogenetic Analysis

**DOI:** 10.3390/insects11090609

**Published:** 2020-09-08

**Authors:** Weijian Huang, Yalin Zhang

**Affiliations:** Key Laboratory of Plant Protection Resources and Pest Management, Ministry of Education, Entomological Museum, College of Plant Protection, Northwest A&F University, Yangling 712100, China; jackyhuang@nwafu.edu.cn

**Keywords:** mitochondrial DNA, leafhopper, Ledrinae, phylogeny

## Abstract

**Simple Summary:**

Ledrinae is a small subfamily with many unique characteristics and comprises 5 tribes with 39 genera including approximately 300 species. The monophyly of Ledrinae and the phylogenetic relationships among cicadellid subfamilies remain controversial. To provide further insight into the taxonomic status and phylogenetic status of Ledrinae, two additional complete mitochondrial genomes of Ledrinae species (*Tituria sagittata* and *Petalocephala chlorophana*) are newly sequenced and comparatively analyzed. The results showed the sequenced genes of Ledrinae retain the putative ancestral order for insects. In this study, phylogenetic analyses based on expanded sampling and gene data from GenBank indicated that Ledrinae appeared as monophyletic with maximum bootstrap support values and maximum Bayesian posterior probabilities. Bayesian inference and maximum likelihood analysis of concatenated alignments of three datasets produced a well-resolved framework of Cicadellidae and valuable data toward future study in this subfamily.

**Abstract:**

Mitochondrial genomes are widely used for investigations into phylogeny, phylogeography, and population genetics. More than 70 mitogenomes have been sequenced for the diverse hemipteran superfamily Membracoidea, but only one partial and two complete mtgenomes mitochondrial genomes have been sequenced for the included subfamily Ledrinae. Here, the complete mitochondrial genomes (mitogenomes) of two additional Ledrinae species are newly sequenced and comparatively analyzed. Results show both mitogenomes are circular, double-stranded molecules, with lengths of 14,927 bp (*Tituria sagittata*) and 14,918 bp (*Petalocephala chlorophana*). The gene order of these two newly sequenced Ledrinae is highly conserved and typical of members of Membracoidea. Similar tandem repeats in the control region were discovered in Ledrinae. Among 13 protein-coding genes (PCGs) of reported Ledrinae mitogenomes, analyses of the sliding window, nucleotide diversity, and nonsynonymous substitution (Ka)/synonymous substitution (Ks) indicate *atp8* is a comparatively fast-evolving gene, while *cox1* is the slowest. Phylogenetic relationships were also reconstructed for the superfamily Membracoidea based on expanded sampling and gene data from GenBank. This study shows that all subfamilies (sensu lato) are recovered as monophyletic. In agreement with previous studies, these results indicate that leafhoppers (Cicadellidae) are paraphyletic with respect to the two recognized families of treehoppers (Aetalionidae and Membracidae). Relationships within Ledrinae were recovered as (*Ledra* + (*Petalocephala* + *Tituria*)).

## 1. Introduction

The typical insect mitochondrial genome is a double-strand circular molecule 15–18 kb in size and contains 37 genes, with 13 protein coding genes (PCGs), two ribosomal RNA (rRNA) genes, and 22 transfer RNA (tRNA) genes [1,2]. During the last ten years, the mitogenome was widely used for studies of phylogeny, phylogeography, and population genetics [2,3,4,5,6,7]. With the reduction in cost of high-throughput sequencing, more than 70 Membracoidea mtgenomes have so far been sequenced and uploaded to pubic databases (Table 1), (https://www.ncbi.nlm.nih.gov). However, they are still underrepresented at the subfamily level, especially in Ledrinae (with only one partial and two complete mtgenomes) [8].

Among the 19 subfamilies of Cicadellidae, Ledrinae is a small group with many unique characteristics and comprises 5 tribes with 39 genera including approximately 300 species [40,41]. The group is cosmopolitan but most diverse in the Old World tropics. Morphologically, ledrines are usually green or brown with the crown and tibiae spatulate, hind femur macrosetal formula of 2 + 1/2 + 0, ocelli on the crown distant from the margin, and the forewing venation usually highly reticulate. Except for members of the Xerophloeini, the Ledrinae are arboreal. According to fossil evidence, ledrines have existed since the Lower Cretaceous [42,43].

The monophyly of Ledrinae and the phylogenetic relationships among cicadellid subfamilies remain controversial. Based on a previous phylogenetic analysis of *28S* rDNA sequences, Dietrich et al. [44] reported the paraphyly of Ledrinae, while a subsequent morphology-based phylogenetic analysis of Ledrinae recovered them as a monophyletic group [9]. A molecular phylogenetic analysis based on DNA sequences from 388 loci and >99,000 aligned nucleotide positions also suggested that Ledrinae is paraphyletic but branches separating the different clades had low support [45]. Recently, a molecular analysis using transcriptomes recovered the included Ledrinae as monophyletic and as a sister group to Tartessinae [46].

As a new tool, mtgenomes may provide further insight into the taxonomic status and phylogenetic status of Ledrinae. Previously, there were only one partial mitogenome of Ledrinae (*Petalocephala ochracea*—KX437734) and two complete ledrine mitogenomes (*Tituria pyramidata*—MN920440 and *Ledra auditura*—MK387845) available in GenBank [8]. In this study, we assembled two mitogenomes (*P. chlorophana* and *T. sagittata*) using next-generation sequencing (NGS) data to provide new evidence toward investigating Ledrinae phylogeny. We report the mitochondrial structure of these two species, including gene order, nucleotide composition, codon usage, tRNA secondary structure, gene overlaps, and the non-coding control region. Using these new sequences, the phylogeny of Membracoidea was reconstructed based on mitogenome information. The objectives of this research were to (1) test the monophyly of Ledrinae and (2) provide a phylogenetic framework for understanding the phylogenetic relationships with other leafhoppers and treehoppers.

## 2. Materials and Methods

### 2.1. Sample Collection and DNA Extraction

Collection information for the adult Ledrinae species sequenced in this study is provided in Appendix A, and specimen identification was based on morphological characters [40]. All specimens were preserved in 100% ethyl alcohol at −20 °C before DNA extraction. The voucher specimens are deposited in the Entomological Museum, Northwest A&F University (NWAFU), Yangling, Shaanxi, China. Total genomic DNA was extracted from the muscle tissues of the thorax using a DNeasy DNA Extraction kit (Qiagen).

### 2.2. Sequencing, Assembly, Annotation and Bioinformatic Analyses

Two complete mtgenomes of Ledrinae were sequenced using next-generation sequencing (Illumina HiSeq 2500; Biomarker Technologies Corporation, Beijing, China). A total of 13,355,846/13,287,460 paired-end clean reads were assembled using Geneious 10.0.5 (Biomatters, Auckland, New Zealand) and MITObim v1.7 software (https://github.com/chrishah/MITObim) [47] with the mitogenome of *Homalodisca vitripennis* (NC006899) and *T. pyramidata* (MN920440) employed as references. The mitogenomes were annotated with Geneious10.0.5 (Biomatters, Auckland, New Zealand). The tRNA genes were identified with MITOS WebServer (http://mitos.bioinf.uni-leipzig.de/index.py) [48] and tRNAscan-SE Search Server v1.21 [49] with the invertebrate mitochondrial genetic code (codon Table 5). The rRNA genes and PCG genes boundaries were determined by the positions of tRNA genes and by alignment with homologous gene sequences and ORF Finder employing codon Table 5 of other leafhoppers. Tandem repeats of the control region (A + T-rich region) were identified with the Tandem Repeats Finder server (http://tandem.bu.edu/trf/trf.html) [50]. The mitogenome map was produced using CGView (http://stothard.afns.ualberta.ca/cgview_server) [51].

The base composition and relative synonymous codon usage (RSCU) of the mitogenomes of Ledrinae species were calculated with MEGA 7.0 (Penn State University, State College, PA, USA) [52] and PhyloSuite v1.2.1 (https://github.com/dongzhang0725/PhyloSuite) [53]. The AT-skew and GC-skew were computed according to the following formulas: AT-skew = [A − T]/[A + T] and GC-skew = [G − C]/[G + C] [54].

The nucleotide diversity (Pi) of PCGs among the Ledrinae species and a sliding window analysis (a sliding window of 200 bp and step size of 20 bp) were conducted using DnaSP 6.0 [55]. DnaSP 6.0 was also used to calculate the rate of non-synonymous (Ka) and synonymous (Ks) substitutions rates for each PCG. Genetic distances between species based on each PCG were estimated using MEGA 7 [52] with Kimura-2-parameter. The genetic distances and Ka/Ks ratios were graphically plotted using Prism 6.01 (GraphPad Software, San Diego, CA, USA). The new mitogenome sequences of Ledrinae (*P. chlorophana* and *T. sagittata*) were registered in GenBank with accession numbers MT610899 and MT610900 (Table 1).

### 2.3. Phylogenetic Analysis

A total of 72 mitogenomes of members of Membracoidea (including 65 leafhoppers, seven treehoppers) were used in our phylogenetic analysis as an ingroup, representing three families and twelve subfamilies. Four species of Fulgoroidea (*Fulgora candelaria*, *Geisha distinctissima*, *Magadhaideus* sp. and *Nilaparvata lugens*), four species of Cercopoidea (*Philaenus spumarius*, *Callitettix braconoides*, *Cosmoscarta dorsimacula* and *Paphnutius ruficeps*) and four species of Cicadoidea (*Magicicada tredecula*, *Diceroprocta semicincta*, *Tettigades auropilosa* and *Tettigarcta crinita*) were selected as outgroups, representing three superfamilies, eight families and twelve subfamilies, respectively. The mitogenomes of two Ledrinae species (*P. chlorophana* and *T. sagittata*) were sequenced in this study while the other sequences of Auchenorrhyncha were obtained from the National Center for Biotechnology Information (NCBI) database (Table 1).

PCGs and RNAs were extracted using PhyloSuite v 1.2.1 [53]. Each PCG was aligned based on codons for amino acids using MAFFT 7 (https://mafft.cbrc.jp/alignment/server/) [56]. All RNAs were aligned with the Q-INS-I algorithm using the MAFFT 7 online service [56]. Ambiguously aligned sites were removed from PCG and RNA alignments using GBlocks v.0.91b (http://molevol.cmima.csic.es/castresana/Gblocks/Gblocks_documentation.html) [57]. Then, all alignments were checked and corrected manually in MEGA 7 [52] for quality. Alignments of all genes were concatenated using PhyloSuite 1.2.1 [53].

Three datasets were generated for phylogenetic reconstruction: (1) PCG123 matrix, which contained all codon positions of the thirteen protein-coding genes (10,668 bp in total); (2) PCG123RNA matrix, which contained all codon positions of the thirteen protein-coding genes and the two rRNA genes (11,535 bp in total); and (3) AA matrix, which contained amino acid sequences of the thirteen protein-coding genes (3272 bp in total). Phylogenetic reconstruction of all matrices was performed using ML (maximum likelihood) and BI (Bayesian inference) analyses, respectively. The best partitioning schemes were selected using PartitionFinder 2.1.1 (www.phylo.org) [58] with the greedy algorithm and BIC criterion (Appendix A). Maximum likelihood analysis was inferred using IQ-TREE [59] under an edge-linked partition model. Branch support analysis was conducted using 10,000 ultrafast bootstrap replicates (UFB) [60]. Bayesian analysis was performed using MrBayes 3.2.6 (www.phylo.org) [61], as implemented in the CIPRES Science Gateway (www.phylo.org) [62]. Each MrBayes analysis involved 10,000,000–20,000,000 generations. The convergence of the independent runs was indicated by a standard deviation of split frequencies <0.01. To mitigate possible effects of base-composition bias and among-site rate heterogeneity, we also used Phylobayes MPI v.1.5a (https://cushion3.sdsc.edu/portal2/login.action;jsessionid=1F85AAEAFBE36B6CCDA96CA816E1A2D3) [63] to analyze the amino acid sequence dataset with the site-heterogeneous model CAT + GTR, as implemented in the CIPRES Science Gateway [62]. We ran two independent tree searches and each run implemented two Markov chain Monte Carlo chains in parallel for at least 30,000 iterations. Runs were terminated when the maxdiff was <0.3 and minimum effective size was >50 (recognized as having reached convergence). A consensus tree was computed from the trees combined from the two runs after the initial 25% trees of each MCMC run were discarded as burn-in.

## 3. Results and Discussion

### 3.1. Genome Organization and Base Composition

In our analysis, we did not find obvious heteroplasty within an individual (almost each site had more than 120 sequences, with more than 90 percent consistency, to determine each base) when we assembled two new complete mtgenomes. The mtgenomes of *P. chlorophana* and *T. sagittata* were 14,927 bp and 14,918 bp long, respectively (Figure 1). Among the four complete mitogenomes of Ledrinae, *T. sagittata* has the smallest mtgenome of 14,198 bp, while *L. auditura* contains the largest at 16,094 bp due to two long non-coding regions [30]. This finding was comparable to the sequence lengths found for other reported Ledrinae mtgenomes [8,30]. Both mitogenomes of Ledrinae included the 37 typical animal mitochondrial genes (13 protein-coding genes, 22 transfer RNA genes, and 2 ribosomal RNA genes) and one non-coding region (A + T-rich region) (Figure 1). The majority strand (J-strand) encoded most of the genes (9 PCGs and 14 tRNAs), while 14 genes (4 PCGs, 8 tRNAs, and 2 rRNAs) were transcribed on the minority strand (N-strand) (Table 2). The gene order of the two newly sequenced Ledrinae was consistent with the typical insect—*Drosophila yakuba* [64]—and other previously-sequenced Ledrinae [8,30].

The overall base composition of *P. chlorophana* was A (29.5%), T (47%), C (9.8%), and G (13.5%) and A (29.8%), T (46.7%), C (10.5%), and G (13%) in *T. sagittata*. Similar to other cicadellid mitogenomes, both mtgenomes were consistently AT nucleotide biased, with 76.6% in *P. chlorophana* and 76.5% in *T. sagittata* (Table 3 and Table 4). The AT nucleotide content of the 13 PCG genes was the lowest (75.3%, 75.5%), while the AT nucleotide content of the A + T-rich region was the highest (85.4%, 82.3%) (Table 3 and Table 4). The composition skew analysis shows a negative AT-skew and a positive GC-skew in the whole mitogenomes (Table 3 and Table 4).

### 3.2. Protein-Coding Genes and Codon Usage

The total size of the 13 PCGs of *P. chlorophana* and *T. sagittata* were 10,911 bp and 10,914 bp, respectively (Table 3 and Table 4). In both sequenced mitogenomes, nine of the 13 PCGs were located on the J-strand, while the other four were oriented on the N-strand (Table 2). Both mitogenomes had almost the same characteristics with the smallest size of *atp8* and the largest size of *nad5* among PCGs. The AT-skews of the PCGs were −0.217 and −0.2 (Table 2).

Most PCGs in the two newly sequenced mitogenomes started with ATN (ATA/T/G/C), except for *nad5* in both newly sequenced Ledrinae, which started with TTG (Table 2), as in the previously sequenced Ledrinae *T. pyramidata* and *P. ochracea* [8], while *nad5* in *L. auditura* started with ATC [30]. Most PCGs of these two complete mitogenomes stopped with a complete termination codon TAA or the incomplete stop codon T, except *nad1* and *nad3* of *P. chlorophana*, which stopped with the termination codon TAG, while TAG was used for *cox1*, *nad3*, and *cytb* in *T. sagittata* (Table 2). In all five Ledrinae mitogenomes, the termination codon TAG occurred less than TAA and at least three incomplete stop codons T were present [30]. Incomplete termination codons are common in insect mitogenomes, which may be related to post-transcriptional modification during the mRNA maturation process [65].

The relative synonymous codon usage (RSCU) values and the amino acid compositions of the four complete Ledrinae mitogenomes are calculated and drawn in Figure 2. The four most frequently used codons were UUU (Phe), UUA (Leu), AUA (Met), and AUU (Ile), which are composed solely of A or U. A distinct preference for using the A or T nucleotides in the third codon positions (Figure 2) reflects the nucleotide A + T bias in the mitochondrial PCGs among Ledrinae.

### 3.3. Transfer and Ribosomal RNA Genes

Twenty-two transfer RNA genes (tRNAs) of *P. chlorophana* and *T. sagittata* mitogenomes are scattered discontinuously over the entire mitogenome (Table 2). The tRNAs region of these two mitogenomes is 1401 bp in *P. chlorophana* and 1398 bp in *T. sagittata*. The AT content of tRNA genes is slightly higher than that of the PCGs, ranging from 77.3% to 77.7% (Table 3 and Table 4). Both positions of 22 tRNA genes were identified in the same relative genomic positions as in *D. yakuba* [64] and previously sequenced Cicadellidae (Table 1), with the exception of three species of Deltocephalinae and one species of Iassinae, which have tRNA rearrangements [18,19,26,53]. The sizes of these 22 tRNAs ranged from 58 bp (*trnS1*) to 72 bp (*trnL2*) in *P. chlorophana* and from 59 bp (*trnE*) to 71 bp (*trnK*) in *T. sagittata* (Table 2). As shown in Figure 3 and Figure 4, all tRNAs can fold into the typical cloverleaf secondary structure, while trnS1 (AGN) formed with the loss of DHU, as recognized in other cicadellid species [18,20,27,29]. The lack of a DHU stem in trnS1 was also commonly present in metazoan mitogenomes [66]. Compared with the two new mitogenomes, we recognize a total of six types of unmatched base pairs (GU, UU, AA, AC, AG, and single A) in the arm structures of tRNAs. A total of 28 weak-bonded GU, 18 mismatched UU, 2 mismatched AA, 2 mismatched AC, and 1 extra single A nucleotide were found in *P. chlorophana* (Figure 4), while 30 mismatched GU, 19 mismatched UU, 1 mismatched UC, 1 mismatched AC, and 1 mismatched AG nucleotide are found in *T. sagittata* (Figure 5). A large number of GU mismatches were also found in other leafhoppers [20,27,31].

Two rRNA genes were encoded from the N-strand in *P. chlorophana* and *T. sagittata*. *rrnL* was 1186/1180 bp (*P. chlorophana*/*T. sagittata*) in length, located in *trnL1* (CUN) and *trnV*, while the small rRNA (*rrnS*) was 734/705 bp (*P. chlorophana*/*T. sagittata*) in length and resided in *trnV* and the control region (Table 2). The lengths range from 1160 bp to 1426 bp in *rrnL* and from 705 bp to 734 bp in *rrnS* in these four complete mitogenomes of Ledrinae [30].

### 3.4. Overlapping Sequences and Intergenic Spacers

A total of 15 gene overlaps occur in the *P. chlorophana* mitogenome with a size from 1 bp to 10 bp, while 13 gene overlaps appear in the *T. sagittata* mitogenome with the same size variation as in the former. The longest overlap region of the two mitogenomes found was 10 bp between nad4 and trnT (Table 2). All four Ledrinae species have one identical overlap region in *trnW*-*trnC* (AAGTCTTA) [30].

There were 8 intergenic spacers in the *P. chlorophana* mtgenome, ranging in size from 1 bp to 23 bp, and the longest two intergenic spacers were 23 bp between *trnQ* and *nad2*. In *T. sagittata*, 8 intergenic spacers were identified, ranging in size from 1 bp to 14 bp, with the longest being between *trnP* and *cytb* (Table 2). There was no identical intergenic spacer region in Ledrinae.

### 3.5. Control Region

In the mtgenome, the A + T-rich region was the longest non-coding sequence. The sizes of the control regions were 694 bp in *P. chlorophana* and 734 bp in *T. sagittata*. In all four complete Ledrinae mitogenomes, all A + T-rich regions were located between *rrnS* and *trnI* and ranged from 694 bp to 856 bp (Figure 5), while *L. auditura* contained two A + T-rich regions. The A + T contents were 85.4% in *P. chlorophana* and 82.3% in *T. sagittata*. Each Ledrinae had specific repeat sequences. *P. chlorophana* had two types of small repeat tandem units with sizes of 18 bp and 43 bp, respectively, while the other three Ledrinae had an absolute tandem repeat ranging from 18 bp to 213 bp (Figure 5). The results indicated that all four complete Ledrinae mitogenome A + T-rich regions had a varied number of absolute tandem repeat units.

### 3.6. Nucleotide Diversity and Evolutionary Rate Analysis

The sliding window analysis exhibited highly variable nucleotide diversity (Pi values) among the 13 aligned PCGs sequences of the four mitogenomes (Figure 6A). The genes *atp8*, *nad6*, *nad5*, and *nad2* had relatively high nucleotide diversities of 0.255, 0.245, 0.231, and 0.230, respectively, while the genes *cox1*, *cytb*, *cox2*, and *cox3* had comparatively low nucleotide diversities of 0.143, 0.161, 0.174, and 0.175 (Figure 6A). Almost the same results were observed according to pairwise genetic distance analysis, with high distances of 0.406, 0.375, 0.342, and 0.341 for *atp8*, *nad6*, *nad5*, and *nad2* and low distances of 0.18, 0.208, 0.233, and 0.233 for *cox1*, *cytb*, *cox3*, and *cox2*, respectively (Figure 6B). The evolutionary rate analysis estimated by the average non-synonymous (Ka) and synonymous (Ks) substitution rates of 13 PCGs among the four mitogenomes ranged from 0.134 to 0.663 (0 < Ka/Ks < 1) (Figure 6B), indicating that these genes are under the purifying selection. The genes *atp8*, *nad6*, and *nad2* showed comparatively high Ka/Ks ratios of 0.663, 0.500, and 0.444, while *cox1*, *cytb*, and *cox2* showed relatively low values of 0.134, 0.184, and 0.188, respectively (Figure 6B).

The mitochondrial gene *cox1* is one of the commonly used barcodes for identifying species and inferring the phylogenetic relationship in leafhoppers [67,68,69], but it was the slowest evolving among the PCGs in these four Ledrinae species, while *atp8* was a comparatively faster evolving gene. In this study, we found *nad6* and *atp8* could be evaluated as potential DNA markers for sibling species delimitation.

### 3.7. Phylogenetic Relationships

The phylogenetic topologies were largely consistent based on analyses of the three datasets (P123, P123R, and AA), with most nodes receiving high support values in BI and ML analyses (Figure 7, Figure 8 and Figure 9). All analyses consistently supported the monophyly of the four superfamilies Fulgoroidea, Cicadoidea, Cercopoidea, and Membracoidea and recovered the relationship (Fulgoroidea + ((Cicadoidea + Cercopoidea) + Membracoidea)) (BS = 100, PP = 1). Within Membracoidea, treehoppers (Membracidae and Aetalionidae) were monophyletic (BS = 100; PP = 1) as a lineage derived from within leafhoppers and as sister to Megophthalminae, consistent with previous studies [8,18,27,44,45,46,47,48,49,50,51,52,53,54,55,56,70]. Within treehoppers, all phylogenetic topologies recovered the relationship (Smiliinae + (Aetalionidae + Centrotinae)) (BS = 100; PP = 1). Meanwhile, Membracoidea was divided into two main groups. Deltocephalinae taxa constituted one clade sister to all remaining Membracoidea, with strong support (BS = 100; PP = 1). This result was congruent with previous results based on mtgenomes and anchored hybrid enrichment data [18,20,27] but different from the results based on *28S* sequences [44] and transcriptomes [46].

Within a paraphyletic Cicadellidae, the cicadellid subfamily Deltocephalinae appeared as monophyletic with maximum bootstrap support values and maximum Bayesian posterior probabilities in all phylogenetic trees, as were Iassinae, Megophthalminae, Typhlocybinae, Coelidiinae, Cicadellinae, Evacanthinae, Eurymelinae, and Ledrinae. However, relationships among these subfamilies varied among analyses and most branches separating them received less than maximum support. Typhlocybinae was sister to all others except Deltocephalinae in BI analyses based on P123R/P123 (PP = 1.00, Figure 7), while Typhlocybinae formed a sister group to Cicadellinae + (Evacanthinae + Ledrinae) in remaining analyses (BS > 92; PP > 0.99, Figure 8 and Figure 9). The relationships among Cicadellinae and Evacanthinae were mostly congruent among results (Figure 6, Figure 7, Figure 8 and Figure 9), with moderate to high support (BS > 90; PP > 0.84). The position of Eurymelinae was inconsistent (Figure 7, Figure 8 and Figure 9) with low to moderate support (BS < 70, PP < 0.86). These results were somewhat discordant with previous molecular phylogenies [44,45,46]. As in other recent phylogenomic analyses of Membracoidea [3,8,18,20,27,46], the deep internal branches separating the subfamilies were very short and received less than maximum support. Most major membracoid lineages appear to have arisen within a very short time interval during the Cretaceous period [43,45,70]. Such ancient rapid radiations often cause problems for phylogenetic reconstruction. In order to mitigate possible effects of base-composition bias and among-site rate heterogeneity, the site-heterogeneous substitution model in Phylobayes was selected, however, the result was almost congruent with a few limited rearrangements among taxa within Deltocephalinae and Ledrinae (Figure 7, Figure 8 and Figure 9).

Within Ledrinae, the five species (*T. pyramidata*, *T. sagittata*, *L. auditura*, *P. ochrace*, and *P. chlorophana*) representing Ledrini formed a monophyletic group with high support values (BS = 100, PP = 1). Expect for Phylobayes analysis topology based on AA, the topology of the remaining analysis (*L. auditura* + ((*P. ochrace* + *P. chlorophana*) + (*T. pyramidata* + *T. sagittata*))) was recovered, with strong support values in BI (PP > 0.97) and low to high support values in ML (BS = 82–100). The monophyly of Ledrinae was consistent with previous studies based on morphological and transcriptome analysis [40,46] but inconsistent with previous analyses using anchored hybrid enrichment genomics and *28S* sequences [44,45]. However, because our analysis only included representatives of one ledrine tribe and taxon sampling in other molecular phylogenetic studies has also been very sparse, data for other tribes will need to be added before a robust test of the monophyly of this subfamily can be performed.

## 4. Conclusions

This study provided the complete mtgenome sequences of *P. chlorophana* and *T. sagittata*, with a comparative analysis of mtgenomes within the subfamily Ledrinae and phylogenetic analysis of the superfamily Membracoidea. The results showed the sequenced genes of Ledrinae retain the putative ancestral order for insects. We suggest that *nad6* and *atp8* show strong potential as DNA markers for species delimitation and phylogenetic relationships among ledrine species. Bayesian inference and maximum likelihood analysis of concatenated alignments of three datasets (P123, P123R, and AA) produced a well-resolved framework, which was largely congruent with previous studies, and with most branches strongly supported except for a few deep internal nodes within Cicadellidae. The present analyses consistently recovered Cicadellidae as paraphyletic, with respect to treehoppers. Within Membracoidea, currently recognized subfamilies for which more than one representative was available were recovered as monophyletic.

The inconsistent results and the short deep internal branches may be due to an ancient rapid radiation (most major cicadellid lineages arose within a *c.* 30 Ma time frame), but limited taxon sampling may also have limited the power of our dataset to resolve relationships among major lineages. These results suggest that mitogenome data are useful for resolving the phylogenetic problem of Cicadellidae, at least at the subfamily level, although this study only included 9 of the 19 currently recognized subfamilies of leafhoppers (sensu 14). Additional mitogenome sampling, especially for representatives of subfamilies and tribes not yet sequenced, may contribute to resolving the phylogeny of Cicadellidae. In addition, limited taxon sampling and single mitogenome may be the main limitations of mitogenomes. In future studies, mtgenomes plus nuclear genes (such as whole *28S*) and additional species would produce a better framework.

## Figures and Tables

**Figure 1 insects-11-00609-f001:**
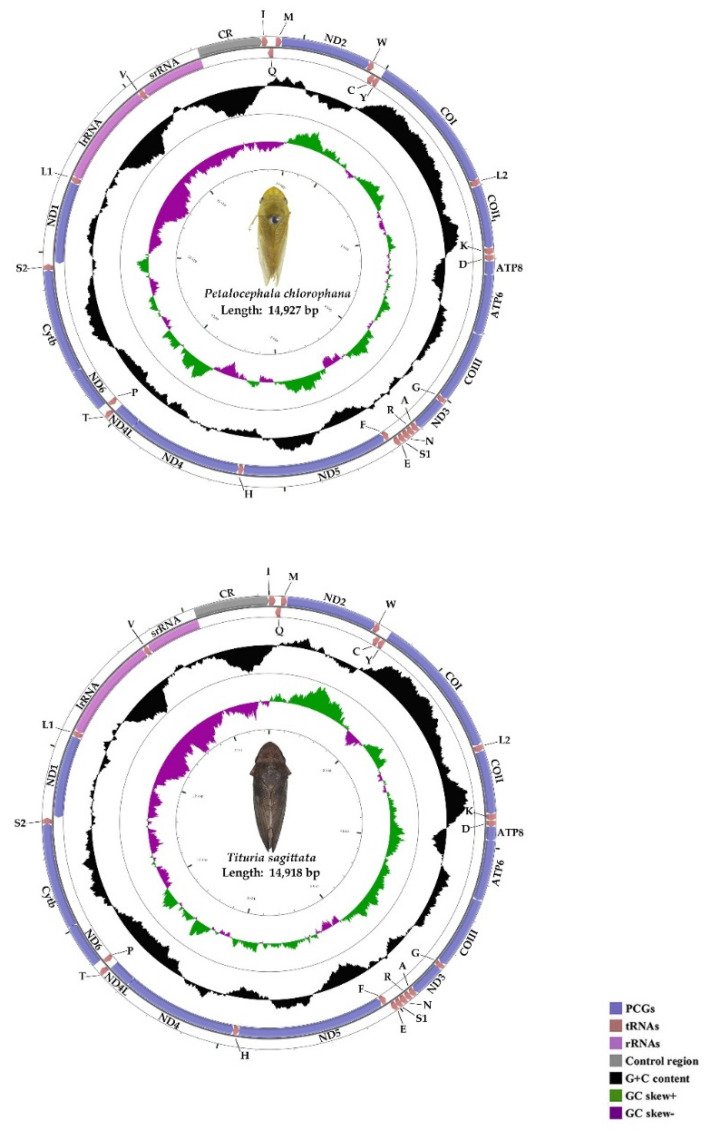
Circular maps of the mtgenomes of *Petalocephala chlorophana* and *Tituria sagittata*.

**Figure 2 insects-11-00609-f002:**
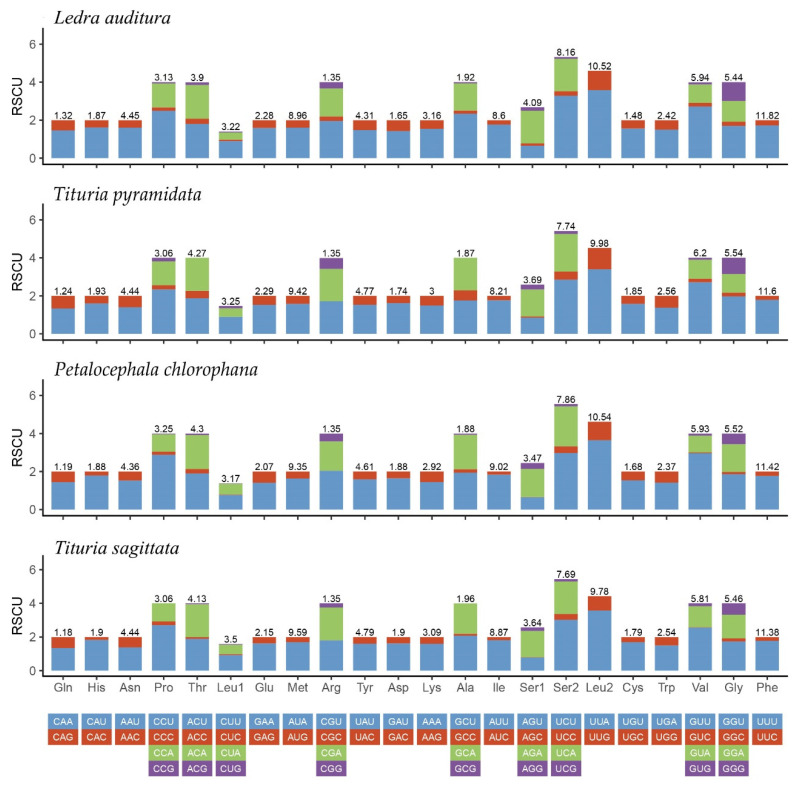
Relative synonymous codon usage (RSCU) of the mitogenomes of four Ledrinae species.

**Figure 3 insects-11-00609-f003:**
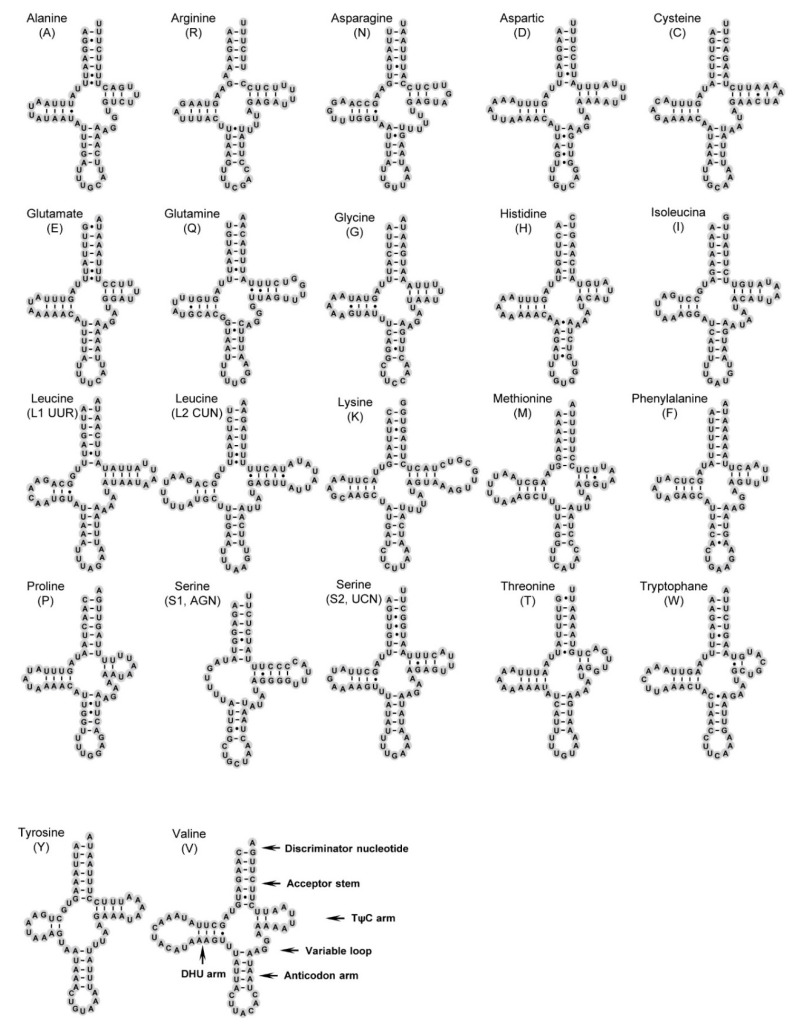
Predicted secondary cloverleaf structure for the tRNAs of *P. chlorophana*.

**Figure 4 insects-11-00609-f004:**
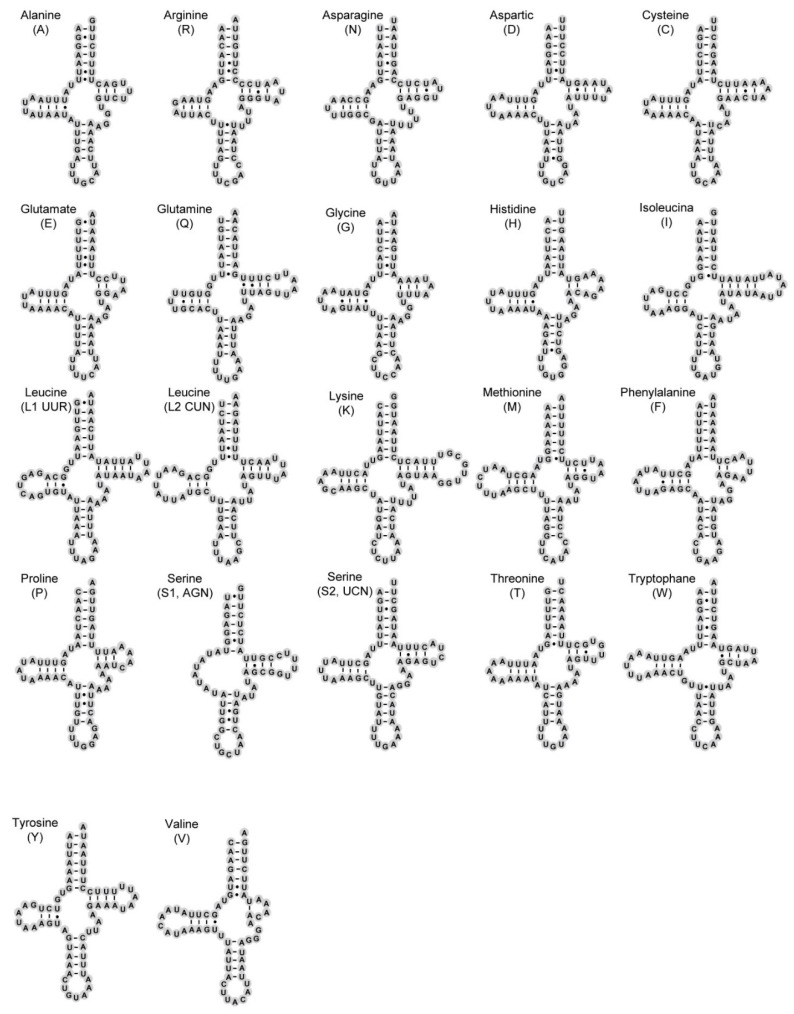
Predicted secondary cloverleaf structure for the tRNAs of *T. sagittata*.

**Figure 5 insects-11-00609-f005:**
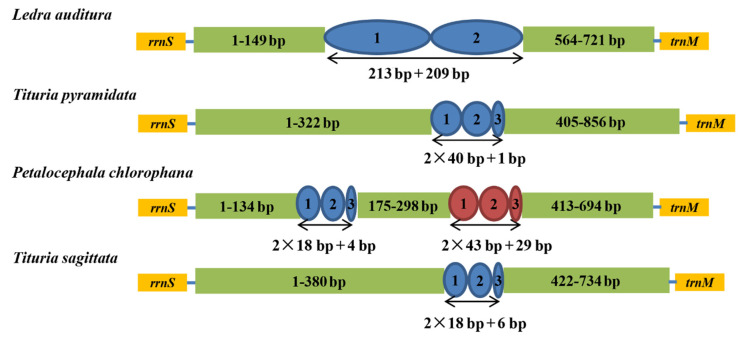
Organization of the control region in Ledrinae mtgenomes. The blue and red ovals indicate the tandem repeats; the non-repeat regions are shown with green boxes.

**Figure 6 insects-11-00609-f006:**
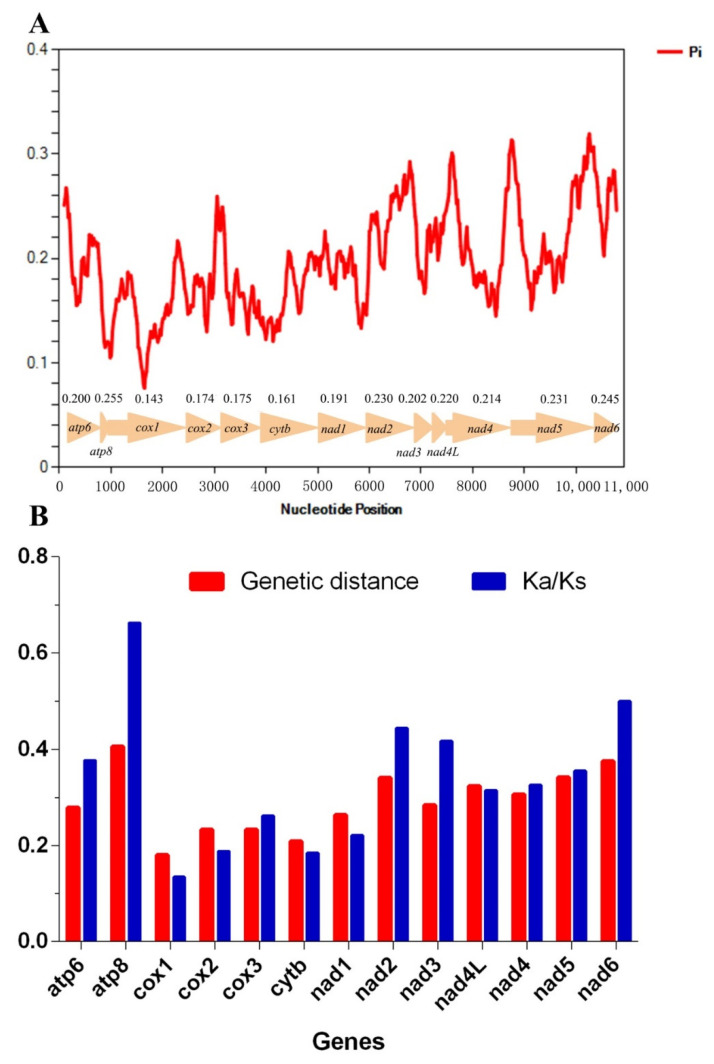
Nucleotide diversities and selection pressures of 13 PCGs in Ledrinae. (**A**) Genetic distance (on average) and ratio of non-synonymous (Ka) to synonymous (Ks) substitution rates of each protein-coding gene among four Ledrinae species. (**B**) Sliding window analysis of protein-coding genes among four Ledrinae species. The red curve shows the value of nucleotide diversity (Pi). Pi value of each PCG is shown above the arrows.

**Figure 7 insects-11-00609-f007:**
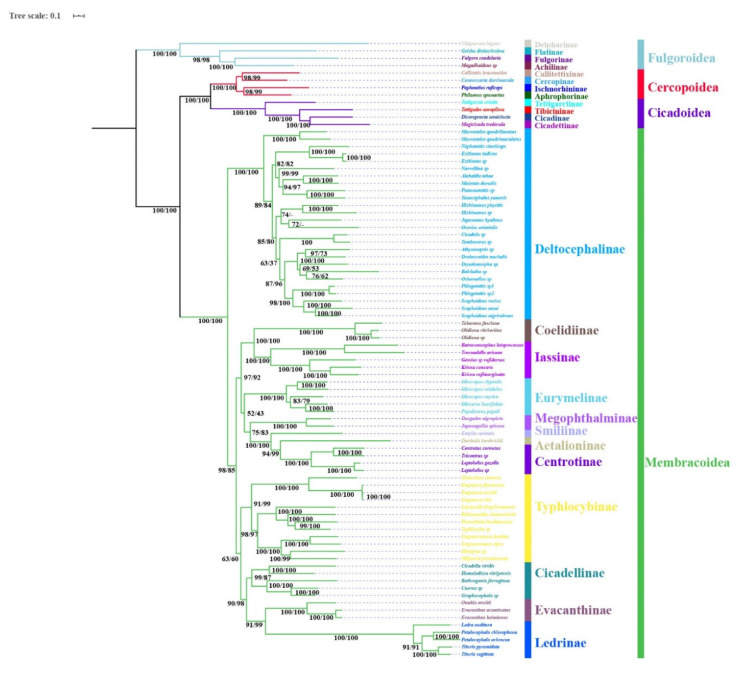
Phylogenetic tree produced using maximum likelihood based on the dataset of P123R/P123. Numerals at nodes are bootstrap support values (BS).

**Figure 8 insects-11-00609-f008:**
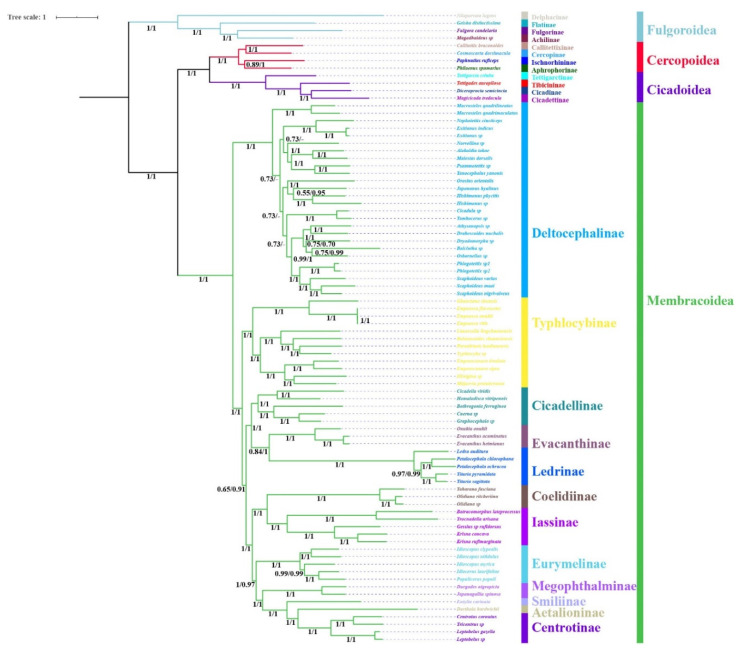
Phylogenetic tree produced using Bayesian inference based on the dataset of P123R/P123. Numerals at nodes are Bayesian posterior probabilities (PP). “-” indicates the clades or species are different in Bayesian inference (BI).

**Figure 9 insects-11-00609-f009:**
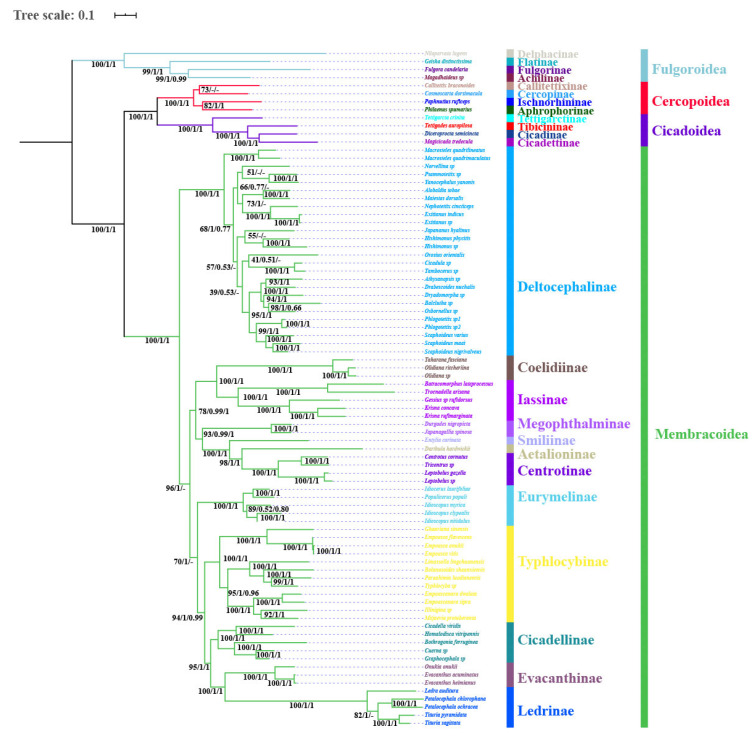
Phylogenetic tree produced using IQtree, Mrbayes and Phylobayes based on the dataset of AA. Numerals at nodes are Bayesian posterior probabilities (PP) and bootstrap support values (BS), respectively. “-” indicates the clades or species are different.

**Table 1 insects-11-00609-t001:** GenBank accession numbers of mtgenomes for species sampled in this study.

Superfamily	Family/Subfamily	Species	Accession Number	Reference
Outgroup				
Fulgoroidea	Fulgoridae/Fulgorinae	*Fulgora candelaria*	NC_019576	[6]
	Flatidae/Flatinae	*Geisha distinctissima*	NC_012617	[7]
	Achilidae/Achilinae	*Magadhaideus* sp.	MH324928	Unpublished
	Delphacidae/Delphacinae	*Nilaparvata lugens*	NC_021748	Unpublished
Cercopoidea	Cercopidae/Cercopinae	*Cosmoscarta dorsimacula*	NC_040115	Unpublished
	Cercopidae/Callitettixinae	*Callitettix braconoides*	NC_025497	[9]
	Cercopidae/Ischnorhininae	*Paphnutius ruficeps*	NC_021100	[10]
	Aphrophoridae/Aphrophorinae	*Philaenus spumarius*	NC_005944	[11]
Cicadoidea	Cicadidae/Cicadinae	*Diceroprocta semicincta*	KM000131	Unpublished
	Cicadidae/Cicadettinae	*Magicicada tredecula*	MH937705	[3]
	Cicadidae/Tibicininae	*Tettigades auropilosa*	KM000129	Unpublished
	Tettigarctidae/Tettigarctinae	*Tettigarcta crinita*	MG737758	[12]
Ingroup				
Membracoidea	Aetalionidae/Aetalioninae	*Darthula hardwickii*	NC_026699	[13]
	Membracidae/Smiliinae	*Entylia carinata*	NC_033539	[14]
	Membracidae/Centrotinae	*Leptobelus gazella*	NC_023219	[15]
	Membracidae/Centrotinae	*Centrotus cornutus*	KX437728	[8]
	Membracidae/Centrotinae	*Tricentrus* sp.	KY039115	[16]
	Membracidae/Centrotinae	*Leptobelus* sp.	JQ910984	[4]
	Cicadellidae/Cicadellinae	*Bothrogonia ferruginea*	KU167550	Unpublished
	Cicadellidae/Cicadellinae	*Cuerna* sp.	KX437741	[8]
	Cicadellidae/Cicadellinae	*Graphocephala* sp.	KX437740	[8]
	Cicadellidae/Cicadellinae	*Cicadella viridis*	KY752061	Unpublished
	Cicadellidae/Cicadellinae	*Homalodisca vitripennis*	NC_006899	Unpublished
	Cicadellidae/Coelidiinae	*Taharana fasciana*	NC_036015	[17]
	Cicadellidae/Coelidiinae	*Olidiana* sp.	KY039119	Unpublished
	Cicadellidae/Coelidiinae	*Olidiana_ritcheriina*	MK738125	Unpublished
	Cicadellidae/Deltocephalinae	*Japananus hyalinus*	NC_036298	[18]
	Cicadellidae/Deltocephalinae	*Maiestas dorsalis*	NC_036296	[18]
	Cicadellidae/Deltocephalinae	*Macrosteles quadrilineatus*	NC_034781	[19]
	Cicadellidae/Deltocephalinae	*Macrosteles quadrimaculatus*	NC_039560	[20]
	Cicadellidae/Deltocephalinae	*Tambocerus* sp.	KT827824	[21]
	Cicadellidae/Deltocephalinae	*Nephotettix cincticeps*	NC_026977	Unpublished
	Cicadellidae/Deltocephalinae	*Hishimonus phycitis*	KX437727	[8]
	Cicadellidae/Deltocephalinae	*Hishimonus* sp.	KX437735	[8]
	Cicadellidae/Deltocephalinae	*Psammotettix* sp.	KX437742	[8]
	Cicadellidae/Deltocephalinae	*Cicadula* sp.	KX437724	[8]
	Cicadellidae/Deltocephalinae	*Exitianus* sp.	KX437722	[8]
	Cicadellidae/Deltocephalinae	*Phlogotettix* sp. 1	KY039135	[16]
	Cicadellidae/Deltocephalinae	*Phlogotettix* sp. 2	KX437721	[8]
	Cicadellidae/Deltocephalinae	*Dryadomorpha* sp.	KX437736	[8]
	Cicadellidae/Deltocephalinae	*Osbornellus* sp.	KX437739	[8]
	Cicadellidae/Deltocephalinae	*Balclutha* sp.	KX437738	[8]
	Cicadellidae/Deltocephalinae	*Scaphoideus varius*	KY817245	[16]
	Cicadellidae/Deltocephalinae	*Scaphoideus nigrivalveus*	KY817244	[16]
	Cicadellidae/Deltocephalinae	*Scaphoideus maai*	KY817243	[16]
	Cicadellidae/Deltocephalinae	*Yanocephalus yanonis*	NC_036131	[16]
	Cicadellidae/Deltocephalinae	*Alobaldia tobae*	KY039116	[16]
	Cicadellidae/Deltocephalinae	*Exitianus indicus*	KY039128	[16]
	Cicadellidae/Deltocephalinae	*Orosius orientalis*	KY039146	[16]
	Cicadellidae/Deltocephalinae	*Athysanopsis* sp.	KX437726	[8]
	Cicadellidae/Deltocephalinae	*Norvellina* sp.	KY039131	[16]
	Cicadellidae/Deltocephalinae	*Drabescoides nuchalis*	NC_028154	[22]
	Cicadellidae/Evacanthinae	*Onukia onukii*	MK251119	[23]
	Cicadellidae/Evacanthinae	*Evacanthus heimianus*	MG813486	[24]
	Cicadellidae/Evacanthinae	*Evacanthus acuminatus*	MK948205	[25]
	Cicadellidae/Iassinae	*Trocnadella arisana*	NC036480	[26]
	Cicadellidae/Iassinae	*Krisna rufimarginata*	NC046068	[26]
	Cicadellidae/Iassinae	*Krisna concava*	MN577635	[26]
	Cicadellidae/Iassinae	*Gessius rufidorsus*	MN577633	[26]
	Cicadellidae/Iassinae	*Batracomorphus lateprocessus*	MG813489	[26]
	Cicadellidae/Eurymelinae	*Populicerus populi*	NC_039427	[27]
	Cicadellidae/Eurymelinae	*Idioscopus nitidulus*	NC_029203	[28]
	Cicadellidae/Eurymelinae	*Idiocerus laurifoliae*	NC_039741	[27]
	Cicadellidae/Eurymelinae	*Idioscopus clypealis*	NC_039642	[29]
	Cicadellidae/Eurymelinae	*Idioscopus myrica*	MH492317	[27]
	Cicadellidae/Ledrinae	*Tituria pyramidata*	MN920440	Unpublished
	Cicadellidae/Ledrinae	*Ledra auditura*	MK387845	[30]
	Cicadellidae/Ledrinae	*Petalocephala ochracea*	KX437734	[8]
	Cicadellidae/Ledrinae	*Petalocephala chlorophana*	MT610899	This study
	Cicadellidae/Ledrinae	*Tituria sagittata*	MT610900	This study
	Cicadellidae/Megophthalminae	*Japanagallia spinosa*	NC_035685	[31]
	Cicadellidae/Megophthalminae	*Durgades nigropicta*	NC_035684	[31]
	Cicadellidae/Typhlocybinae	*Illinigina* sp.	KY039129	[16]
	Cicadellidae/Typhlocybinae	*Empoasca onukii*	NC_037210	[32]
	Cicadellidae/Typhlocybinae	*Empoasca vitis*	NC_024838	[33]
	Cicadellidae/Typhlocybinae	*Typhlocyba* sp.	KY039138	[16]
	Cicadellidae/Typhlocybinae	*Empoascanara dwalata*	MT350235	Unpublished
	Cicadellidae/Typhlocybinae	*Empoasca flavescens*	MK211224	[34]
	Cicadellidae/Typhlocybinae	*Bolanusoides shaanxiensis*	MN661136	Unpublished
	Cicadellidae/Typhlocybinae	*Limassolla lingchuanensis*	NC046037	[35]
	Cicadellidae/Typhlocybinae	*Empoascanara sipra*	MN604278	[36]
	Cicadellidae/Typhlocybinae	*Paraahimia luodianensis*	NC047464	[37]
	Cicadellidae/Typhlocybinae	*Mitjaevia protuberanta*	NC047465	[38]
	Cicadellidae/Typhlocybinae	*Ghauriana sinensis*	MN699874	[39]

**Table 2 insects-11-00609-t002:** Organization of the mtgenomes of *P. chlorophana* and *T. sagittata.*

Name	Location		Size (bp)	Intergenic	Codon		Strand
	From	To		Nucleotides	Start	Stop	
*trnI*	1/1	64/69	64/69				J/J
*trnQ*	62/67	126/130	65/64	−3/−3			N/N
*trnM*	150/130	215/194	65/65	23/−1			J/J
*nad2*	215/195	1189/1169	975/975	-/-	ATT/ATT	TAA/TAA	J/J
*trnW*	1188/1169	1250/1232	63/64	−2/−1			J/J
*trnC*	1243/1225	1307/1288	65/64	−8/−1			N/N
*trnY*	1307/1292	1370/1354	64/63	−1/3			N/N
*cox1*	1372/1354	2910/2889	1539/1536	1/−1	ATG/ATG	TAA/TAG	J/J
*trnL2*(UUR)	2906/2892	2977/2960	72/69	−5/2			J/J
*cox2*	2978/2961	3656/3639	679/679	-/-	ATT/ATT	T/T	J/J
*trnK*	3657/3640	3727/3710	71/71	-/-			J/J
*trnD*	3731/3713	3796/3775	66/63	3/2			J/J
*atp8*	3797/3776	3949/3928	153/153	-/-	ATT/ATT	TAA/TAA	J/J
*atp6*	3946/3925	4599/4578	654/654	−4/−4	ATG/ATA	TAA/TAA	J/J
*cox3*	4600/4580	5379/5357	780/778	-/1	ATG/ATG	TAA/T	J/J
*trnG*	5379/5358	5439/5418	61/61	−1/-			J/J
*nad3*	5437/5416	5793/5772	357/357	−3/−3	ATA/ATA	TAG/TAG	J/J
*trnA*	5797/5771	5856/5830	60/60	3/−2			J/J
*trnR*	5857/5833	5917/5892	61/60	-/2			J/J
*trnN*	5917/5894	5980/5957	64/64	−1/1			J/J
*trnS1*(ACN)	5981/5957	6038/6016	58/60	-/−1			J/J
*trnE*	6040/6016	6099/6074	60/59	1/−1			J/J
*trnF*	6099/6073	6160/6136	62/64	−2/−2			N/N
*nad5*	6160/6137	7819/7799	1660/1663	−1/-	TTG/TTG	T/T	N/N
*trnH*	7820/7800	7880/7861	61/62	-/-			N/N
*nad4*	7880/7861	9169/9156	1290/1296	−1/−1	ATC/ATC	TAA/TAA	N/N
*nad4L*	9160/9147	9498/9487	276/276	−10/−10	ATG/ATG	TAA/TAAS	N/N
*trnT*	9438/9425	9498/9487	61/63	2/2			J/J
*trnP*	9499/9488	9559/9549	61/62	-/-			N/N
*nad6*	9574/9564	10,056/10,049	483/486	14/14	ATT/ATT	TAA/TAA	J/J
*cytb*	10,049/10,042	11,185/11,178	1137/1137	−8/−8	ATG/ATG	TAA/TAG	J/J
*trnS2*(UCN)	11,185/11,177	11,248/11,240	64/64	−1/−2			J/J
*nad1*	11,251/11,245	12,180/12,172	930/928	2/4	ATA/ATA	TAG/T	N/N
*trnL1*(CUN)	12,181/12,173	12,245/12,238	65/66	-/-			N/N
*rrnL*	12,246/12,239	13,431/13,418	1186/1180	-/-			N/N
*trnV*	13,432/13,419	13,499/13,478	68/60	-/-			N/N
*rrnS*	13,500/13,479	14,233/14,183	734/705	-/-			N/N
*CR*	14,234/14,184	14,927/14,917	694/734	-/-			J/J

**Table 3 insects-11-00609-t003:** Nucleotide composition and skewness comparison of different elements of the *Petalocephala chlorophana* mtgenome.

Feature	Length (bp)	T%	C%	A%	G%	A + T%	AT-Skew	GC-Skew
PCGs	10,911	45.8	11.4	29.5	13.3	75.3	−0.217	0.076
Control region	694	43.2	6.6	42.2	7.9	85.4	−0.012	0.089
tRNAs	1401	40.8	9.1	36.5	13.6	77.3	−0.054	0.195
rRNAs	1920	34.5	8.9	46.1	10.5	80.6	0.145	0.081
Whole genome	14,927	47	9.8	29.6	13.5	76.6	−0.227	0.157

**Table 4 insects-11-00609-t004:** Nucleotide composition and skewness comparison of different elements of the *Tituria sagittata* mtgenome.

Feature	Length (bp)	T%	C%	A%	G%	A + T%	AT-Skew	GC-Skew
PCGs	10,914	45.3	11.4	30.2	13.1	75.5	−0.2	0.068
Control region	734	43.1	8.9	39.2	8.9	82.3	−0.046	0
tRNAs	1398	41.6	8.9	36.1	13.4	77.7	−0.071	0.203
rRNAs	1885	34.2	9.2	44.9	11.7	79.1	0.136	0.117
Whole genome	14,918	46.7	10.5	29.8	13	76.5	−0.222	0.109

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
