# Peer review of "Characterization of Two Complete Mitochondrial Genomes of Ledrinae (Hemiptera: Cicadellidae) and Phylogenetic Analysis"

_insects, 2020, doi:10.3390/insects11090609_

Round 1

Reviewer 1 Report

I have carefully read the submission entitled ‘Characterization of two complete mitochondrial genomes of Ledrinae (Hemiptera: Cicadellidae) and phylogenetic analysis’ by Weijian Huang and Yalin Zhang.

The authors present two new complete mitochondrial genomes of two species of the Ledrinae family, which has been rather underrepresented in respect to complete mitochondrial genomes. In this respect, a gap has been identified and this research is considered as a novel contribution in the field.

This submission is well-written, and analysis seems to have been properly performed. Previous research in the field has been properly acknowledged. Below you can find some comments that, in my opinion, would help the authors to present a more ‘complete’ submission of broader interest.

Major comments:

  1. Introduction and discussion: as I can see, the authors base their analysis on a single individual per species and a single molecule (mitogenome). Advantages of mitogenomes are well discussed but I would like to see the presentation of the limitations when mitochondrial genes are used as stand-alone tools for phylogenetic analysis (including, for example, differential evolution compared to the chromosomal genome, mitochondrial sweeps originating from different factors such as symbionts and hybridization, and others).
  2. High throughput sequencing can help identify levels of heteroplasmy within an individual with respect to mutations of the mitogenome. The authors do not seem to have taken this into account. It would be nice to include such an analysis or at least discuss this possibility and clearly present how they addressed possibly polymorphic positions.

Minor comments

  1. Line 104 (and elsewhere): numbers until up to ten (and even more) should be written in full (four instead of 4 and so on).
  2. L262: a comparatively faster evolving
  3. L262-263: please rewrite this sentence since it is not clear.
  4. L290-291: you can omit ‘in our analysis’. This is obvious by the sentence itself.
  5. L300: ‘Expect’ or ‘Except for’?

Author Response

Response to Reviewer 1 Comments

The manuscript "Characterization of two complete mitochondrial genomes of Ledrinae (Hemiptera: Cicadellidae) and phylogenetic analysis "presents important scientific information that contributes to the evolutionary knowledge of hemipterans. Before publication, I suggest only the following modifications:

  • After the first abbreviation for mitochondrial genome (line 32), change the entire manuscript to mtgenome.

Response: Accepted and changed mitochondrial genome of the entire manuscript to mtgenome.

  • Line 59: Change " (Petalocephala chlorophana and Tituria sagittata)" for " ( chlorophana and T. sagittata)"

Response: Accepted and used the abbreviation. On line 59

  • Line 80: Change "Tituria pyramidata" for " pyramidata

Response: Accepted and used the abbreviation. On line 80

  • Line 99: (P. chlorophana and T. sagittata) must be presented in italics.

Response: Accepted and used italics. On line 99

  • Line 195: Change " Drosophila yakuba Burla" for " yakuba"

Response: Accepted and used the abbreviation. On line 199

Reviewer 2 Report

The manuscript "Characterization of two complete mitochondrial genomes of Ledrinae (Hemiptera: Cicadellidae) and phylogenetic analysis "presents important scientific information that contributes to the evolutionary knowledge of hemipterans. Before publication, I suggest only the following modifications:

1) After the first abbreviation for mitochondrial genome (line 32), change the entire manuscript to mtgenome.

2) Line 59: Change " (Petalocephala chlorophana and Tituria sagittata)" for " (P. chlorophana and T. sagittata)"

3) Line 80: Change "Tituria pyramidata" for "T. pyramidata"

4) Line 99: (P. chlorophana and T. sagittata) must be presented in italics.

5) Line 195: Change " Drosophila yakuba Burla" for " D. yakuba"

Author Response

Response to Reviewer 2 Comments

I have carefully read the submission entitled ‘Characterization of two complete mitochondrial genomes of Ledrinae (Hemiptera: Cicadellidae) and phylogenetic analysis’ by Weijian Huang and Yalin Zhang.

The authors present two new complete mitochondrial genomes of two species of the Ledrinae family, which has been rather underrepresented in respect to complete mitochondrial genomes. In this respect, a gap has been identified and this research is considered as a novel contribution in the field.

This submission is well-written, and analysis seems to have been properly performed. Previous research in the field has been properly acknowledged. Below you can find some comments that, in my opinion, would help the authors to present a more ‘complete’ submission of broader interest.

Major comments:

Introduction and discussion: as I can see, the authors base their analysis on a single individual per species and a single molecule (mitogenome). Advantages of mitogenomes are well discussed but I would like to see the presentation of the limitations when mitochondrial genes are used as stand-alone tools for phylogenetic analysis (including, for example, differential evolution compared to the chromosomal genome, mitochondrial sweeps originating from different factors such as symbionts and hybridization, and others).

Response: As we are finding, even with very large anchored hybrid (Dietrich, 2017) and transcriptome (Skinner, 2020) datasets (contain chromosomal genes), relationships among subfamilies of Cicadellidae are very difficult to resolve, and even branches with 100% boostrap support can be unstable. Because most of the major cicadellid lineages appeared almost at the same time during the Cretaceous, giving rise to very short, deep internal branches, it is possible that we may never be able to resolve some relationships satisfactorily, even with very large datasets.

Some researchers believe mitochondria originate from different factors such as symbionts and hybridization, and others. We still believe mtgenomes originate from some kind of biofilm of the cell, at least, it is a part of a cell now.

As Stephen L. Cameron (2014) suggested, ’ The small size of the mtgenome makes it a practical genome study system that will not be equaled by nuclear genome sequencing in the near future.’ In conclusions, we suggested limited taxon sampling and single mitogenome may be the main limitations of mitogenomes. So, we added follow sentences: In addition, limited taxon sampling and single mitogenome may be the main limitations of mitogenomes. In future studies, mtgenomes plus nuclear genes (such as whole 28S) and additional species would a better famework. On line 345-347

High throughput sequencing can help identify levels of heteroplasmy within an individual with respect to mutations of the mitogenome. The authors do not seem to have taken this into account. It would be nice to include such an analysis or at least discuss this possibility and clearly present how they addressed possibly polymorphic positions.

Response: There is heteroplasmy within an individual or some species with respect to mutations of the mitogenome. In our analysis, we didn’t find obvious heteroplasmy (Almost each site has more than 120 sequences, with more than 90 percent consistency, to determine the base) when we assembled two new complete mtgenomes. In addition, we fund some mtgenomes (download from genebank) containing degenerate bases, usually in A + T-rich region, and the authors just left it in annotation. This may due to that control region sequences are not used in the reconstruction of phylogenetic trees.

So, we added follow sentence: In our analysis, we didn’t find obvious heteroplasmy within an individual (Almost each site had more than 120 sequences, with more than 90 percent consistency, to determine each base) when we assembled two new complete mtgenomes. On line 142-144

Minor comments

  • Line 104 (and elsewhere): numbers until up to ten (and even more) should be written in full (four instead of 4 and so on).

Response: Accepted and rewritten all numbers (less than ten) in full. On line 102- 108

  • L262: a comparatively faster evolving

Response: Accepted and changed. On line 265

  • L262-263: please rewrite this sentence since it is not clear.

Response: Accepted and rewritten as follow:

In this study, we found nad6 and atp8 would be evaluated as potential DNA markers for sibling species delimitation. On line 267

  • L290-291: you can omit ‘in our analysis’. This is obvious by the sentence itself.

Response: Accepted and omitted ‘in our analysis’. On line 294-295

  • L300: ‘Expect’ or ‘Except for’?

Response: Accepted and used ‘Except for’ instead to ‘Expect’. On line 304